# Microstructure and Mechanical Properties of Wire Arc Additively Manufactured MoNbTaWTi High Entropy Alloys

**DOI:** 10.3390/ma14164512

**Published:** 2021-08-11

**Authors:** Jian Liu, Jing Li, Xian Du, Yonggang Tong, Rui Wang, Dongyu He, Zhihai Cai, Haidou Wang

**Affiliations:** 1National Engineering Research Center for Remanufacturing, Army Academy of Armored Forces, Beijing 100072, China; xbdliu5899@163.com (J.L.); gzulijing@163.com (J.L.); zlbdy@163.com (X.D.); dzpywangrui@163.com (R.W.); wanghaidou@tsinghua.org.cn (H.W.); 2College of Automotive and Mechanical Engineering, Changsha University of Science & Technology, Changsha 410000, China; tongyonggang_csust@163.com; 3National Key Laboratory for Remanufacturing, Army Academy of Armored Forces, Beijing 100072, China; hedongyu116@163.com

**Keywords:** wire arc additive manufacturing, HEAs, MoNbTaWTi, microstructure and properties

## Abstract

High-temperature resistant high-entropy alloys (HEAs) have attracted extensive attention due to their excellent thermodynamic stability and mechanical properties, especially at high temperatures. However, a highly effective method for large-size HEAs is still desirable but challengeable. This research reported a facile yet effective strategy for MoNbTaWTi HEAs via in-situ wire arc additive manufacturing (WAAM). The wire was MoNbTaWTi cable-type welding wire (CTWW) consisting of one center wire and seven twisted peripheral wires. Then, additive manufacturing of MoNbTaWTi high entropy alloys (HEAs) was accomplished, and various analytical techniques studied the microstructures and mechanical properties of the overlaying formed layers. X-ray diffraction showed the overlaying formed layers to contain a single disordered BCC solid solution phase with high-temperature structural stability. In addition, the single-phase BCC structure was maintained from 0 to 1400 °C. The bottom of the overlaying formed layers was made of columnar cellular structure, and the upper part resembled “cauliflower-like” fine dendrite and equiaxed crystal structure. The hardness of the overlaying formed layers averaged 533 HV_0.2_ at room temperature. At 1000 °C, the hardness was around 110 HV_1_, close to the value of Inconel 718 alloy (125 HV_1_). The compressive strength of the overlaying formed alloy layers displayed no sensitivity towards change in temperature from 500 to 1000 °C. As the temperature rose from 500 to 1000 °C, the compressive strength changed from 629 to 602 MPa, equivalent to only a 27 MPa decrease. The latter was much higher than the strength of Inconel 718 alloy at the same temperature (200 MPa).

## 1. Introduction

As a novel type of alloy including at least five principal elements, high entropy alloys (HEAs) have attracted extensive attention in science or industry since Prof. Yeh [1] and Prof. Cantor [2] proposed this concept separately and simultaneously in 2004. The high entropy thermodynamics, slow diffusion dynamics, lattice structural distortion, and cocktail performance [3]. Thus, it could easily form simple face-centered cubic (FCC) or body-centered cubic (BCC) solid solutions and nanoscale precipitates, leading to better properties in terms of high strength, elevated hardness, excellent thermal stability, good corrosion, and wear resistance.

In the past nearly two decades, numerous HEAs such as light-weight high entropy alloys (LWHEAs), high-temperature resistant HEAs, radiation resistance HEAs, and soft magnetic HEAs have so far been developed, with ingots, bares, powders, coatings, and films material patterns, by using vacuum arc melting [4,5,6,7,8,9], powder metallurgy [10,11,12], magnetron sputtering [13,14], mechanical alloying [15], spark plasma sintering [16,17], and laser cladding [18,19,20] methods. Especially, high-temperature resistant HEAs have achieved extensive interest, with the increasing booming of the aeroengine, nuclear power plant, machining tools, gas turbines recently. For instance, Su et al. used the vacuum arc melting method and prepared high melting point alloys with equal molar ratios, such as TiZrHfVNb and TiZrHfVTa, TiZrHfNbMo, and TiZrVHfMo. The study of microstructures and properties of these alloys revealed as-cast alloys with phases of BCC, HCP, or BCC+ HCP solid solution. Laves phase also existed in TiZrVHfMo. The hardness of as-developed alloys ranged between 334 HB and 522 HB, and compress strength varied from 1663 MPa to 2060 MPa [4]. In 2010, Senkov et al. [20,21] prepared a new alloy, published as“Refractory High Entropy Alloys” in the “Intermetallics” periodical for the first time. Lilensten et al. successfully developed Ti35Zr27.5Hf27.5Ta5Nb5 alloy with a rhombic structure at different molar ratios [22]. Given the insufficient strength of HfNbTaTiZr alloy at high temperature, Chien et al. [23] designed and successfully synthesized HfMoTaTiZr and HfMoNbTaTiZr alloys with BCC single-phase structure. The as-developed alloys showed superior properties to those of HfNbTaTiZr, with fracture strain of HfMoNbTaTiZr reaching 12% at room temperature. Huang et al. [24] reported TiZrHfTaX refractory HEAs with good strength and plasticity. According to transformation-induced plasticity (TRIP) and reduced Ta content, the structure of these alloys changed from BCC single phase to (BCC+HCP) dual-phase. In 2018, Lv et al. [25] added certain amounts of oxygen into the TiZrHfNb model alloy, leading to the formation of a new state of an interstitial atom in the lattice named as ordered interstitial atom complex. Subsequently, high yield strength and elongation up to about 1.11 GPa and 27.66% respectively were exhibited due to the presence of an ordered oxygen complex. Besides, many other researchers, such as Jayaraj [9], SaadSheik [26], Zhang Yong [27], Tong [28], etc., have developed lots of studies on high-temperature resistant HEAs.

However, most of the research objects of HEAs were small size ingots or powder at present. No mature technology has been employed to prepare large-size HEAs with high efficiency for engineering applications, using traditional fusion cast methods due to the macrosegregation bottleneck for large-size HEAs fabrication.

Wire arc surfacing is an additive manufacturing technology with high efficiency, elevated material utilization, and simple equipment. As the small molten pool effect, the macrosegregation bottleneck can be prevented, thereby beneficial for engineering preparation of large-size high-temperature resistant HEAs and their applications. However, high-temperature resistant HEAs wire obtaining is still desirable but challengeable. In this study, MoNbTaWTi HEAs cable-type welding wire (CTWW) with a nominal composition of Mo_30.7_Nb_13.4_Ta_13.4_W_15.2_Ti_27.3_ was created to provide a solution for large-size high-temperature resistant HEAs preparation. Then, the forming experiment of MoNbTaWTi HEAs by WAAM was developed, and various analytical techniques analyzed the microstructures and mechanical properties of the formed layers.

## 2. Experimental

### Procedures

Ti-6Al-4V alloy (Provided by Baoji FuTaiyuan Metal Materials Co. Ltd., Baoji, China) with a size of 100 mm × 100 mm × 10 mm was used as the substrate. The oxide skin was firstly dislodged using a hand-held grinding wheel grinder until the surface showed a metallic luster. Then, it was cleaned using ultrasonic acetone solution and wiped with alcohol before being stored in an oven to dry.

The MoNbTaWTi HEAs cable-type welding wire (CTWW) was twisted stranded from seven branch wires with a diameter of 0.5 mm using special twisted stranding equipment. The stranding distance was set to 12 mm. The purity of each branch wire was not less than 99.9%. The schematic for the CTWW and prepared wire graph was shown in Figure 1.

The forming process experiment was performed at a feeding speed of 4.5 m/min, surfacing speeding of 0.3 m/min, shielding gas flow of 10 L/min, surfacing current of 280 A, and surfacing voltage of 29 V, after pretreatment process on the substrate, via Pulse MIG- 500II (KEMPPI Company, Lahti, Finland). A self-made argon shield case was employed to prevent oxidation during experimentation. The ambient temperature oscillated around 20 °C.

The microstructure and morphology analysis was carried out on a hot inlay sample, after corroded for 30 s in mixed hydrofluoric acid, nitric acid, and water at a volume ratio of 1:1:1, using optical microscopy (OM, OLYMPUS-DP12, Olympus Corporation, Tokyo, Japan), scanning electron microscopy (SEM, Quanta 200, FEI Co., Hillsboro, GA, USA), energy dispersive spectrometry (EDS, X-Max80, Oxford Instruments Co., Oxford, UK) and X-ray diffraction (XRD, D8Advance, Bruker AG, Leipzig, Germany).

METTLER TOLEDO TGA/DSC synchronous thermal analyzer (Mettler Toledo Co., Zurich, Switzerland) was used for phase transition point analysis. The heating rate was set to 15 °C/min, and the highest temperature was limited to 1400 °C.

The hardness distribution was tested by Vickers hardness test equipment (MICROMET-6030, Buehler Co., Lake Bluff, IL, USA) at a load of 200 g and load holding time of 20 s. The height direction of the surfacing layer was measured every 0.2 mm along the top center of the surfacing layer until reaching the substrate. The high-temperature hardness was tested at 1000 °C by the HTV-PHS30 system (Archimedes Industrial Technology Co., Ltd., London, UK), with a heating speed of 60 K/min, a measuring load of 100 g, and a load holding time of 10 s. For comparison, the hardness of Inconel 718 superalloy at 1000 °C was also measured as a control.

The high-temperature compression test temperatures were set to 500 °C and 1000 °C using GLEEBLE-3500 thermal simulation test machine (DATA SCIENCES INTERNATIONAL, INC., St. Paul, MN, USA) with a sample size of φ2 × 4 mm, compression rate of 10^−^^3^s^−^^1^, compression amount of 40%, a heating rate of 100 °C/min, and holding time of 5 min.

## 3. Results

### 3.1. Microstructure and Morphology

Figure 2 shows the XRD pattern of the surfacing formed layer. MoNbTaWTi alloy displayed a single-phase BCC solid solution with no ordered peak at low diffraction angles, indicating the successful formation of a single disordered BCC solid solution phase.

The lattice constant of the phase structure of surfacing formed MoNbTaWTi alloy was estimated at 321.23 pm and calculated at 322.69 pm. Thus, it could be concluded that surfacing formed MoNbTaWTi alloy fitted well with the above mixing principle, confirming the disordered solid solution phase of the formed layer.

The microstructures of the bottom and middle parts of the formed MoNbTaWTi layer are illustrated in Figure 3. The bottom structure of the surfacing layer was mainly composed of directional epitaxy-grown cellular crystals (Figure 3a), which gradually changed into columnar dendrite structure as solidification further proceeded. In the middle and upper parts of the surfacing layer, the dendrite was first subjected to impact, and then broking and re-melting occurred due to the weakening of heat flow direction and disturbance of arc in the molten pool. This led to the formation of “cauliflower-like” finer dendrite and equiaxed crystal structure (Figure 3b).

The alloying elements of surfacing formed layers looked uniformly distributed without obvious component segregation. However, the micro-scaled area-scanning results of distributed elements displayed concentrated W and Ta elements in the dendrites (Figure 4). By comparison, the Ti element was mostly distributed at the intergranular, and both Nb and Mo were evenly distributed in both regions. Moreover, this means that certain dendrite segregation existed. Table 1 provides each element’s segregation rate (SR), defined as the ratio of elements in intragranular to intergranular phases. Note that more deviation between SR and 1 led to more serious dendrite segregation.

### 3.2. Hardness of WNbMoTaTi Surfacing Layer

The distribution curve of Vickers hardness of the formed MoNbTaWTi alloy layer is displayed in Figure 5. The overall hardness of the surfacing formed layer averaged 533 HV_0.2_.

The hardness of the surfacing formed layer appeared much higher than that of the Ti-6Al-4V alloy. According to the literature [21], the hardness of WNbMoTa prepared by the vacuum arc melting method was only 480 HV_0.2_. Therefore, the hardness of high entropy alloy formed by overlaying technology looked higher. Since both prepared samples had a single BCC structure, the high hardness of obtained overlaying layer should be attributed to the relative refinement of its grains.

Figure 6 compares the hardness of overlaying layer of MoNbTaWTi refractory HEA with that of Inconel 718 nickel-base superalloy at 1000 °C. The average hardness of the overlaying layer at 1000 °C was estimated to be about 110 HV_1_, which was close to the 125 HV_1_ of Inconel 718 alloy.

### 3.3. Compression Performance of WNbMoTaTi Surfacing Layer

The DSC test result showed the melting point of the overlaying layer was above 1400 °C, as shown in Figure 7, and no phase change in overlaying layer from 0 to 1400 °C but single-phase BCC structure was preserved. The highest melting point of Inconel 718 was reported as 1300 °C [7]. Thus, MoNbTaWTi refractory HEAs prepared by overlaying technology possessed elevated temperature softening resistance and good thermal stability.

The compressive stress-strain curves of the overlaying layers of MoNbTaWTi refractory HEAs at 500 °C and 1000 °C are presented in Figure 8a,b. At 500 °C, the compressive strength of the surfacing formed layer was recorded as 629 MPa, and the fracture strain was 2.1%. At 1000 °C, the surface-formed layer’s compressive strength was 602 MPa, which was only 27 MPa lower than that at 500 °C. The fracture strain was estimated to be 4.9%, which was 1.33-fold higher than that at 500 °C. Hence, the increase in temperature had little effect on the surface-formed layer’s compressive strength from 500 to 1000 °C but can greatly improve the plastic toughness. On the other hand, the surfacing formed layer displayed high temperature softening resistance. By comparison, the compressive strength of Inconel 718 alloy at 1000 °C was reported as only 200 MPa, meaning that the high-temperature strength of the surfacing formed layer of WNbMoTaTi HEA was far superior to that of Inconel 718 alloy.

However, the compressive stress-strain curve of the surfacing formed layer of WNbMoTaTi HEA at 500 °C looked relatively simple compared to the curves exhibited in Figure 8a,b, where stress rose continually with strain until fracture occurred. By contrast, two obvious steps were noticed in the middle of the surface-formed layer’s compressive stress-strain curve at 1000 °C. The compression deformation can be divided into three stages. First, stress rose rapidly with strain then decreased before becoming stable. Next, stress increased continuously with strain and then declined instantaneously after reaching a certain value. Afterward, the increase continued until the occurrence of fracture.

The fracture morphology analysis could shed light on the fracture mechanism. The fracture morphologies of surfacing formed layers of WNbMoTaTi HEAs at 500 °C and 1000 °C are presented in Figure 9a,b. The fracture surface at 500 °C mainly displayed cleavage steps in addition to obvious cracks. Therefore, the compression fracture of the surfacing formed layer of WNbMoTaTi HEA at 500 °C was identified as brittle cleavage fracture. At 1000 °C, the proportion of cleavage steps decreased, and small dimples with some tearing edges appeared. Therefore, the fracture mode of surfacing formed layer at 1000 °C changed into composite fracture mode of ductile fracture + quasi cleavage fracture. This meant that the plastic toughness of the surfacing formed layer of WNbMoTaTi HEA at 1000 °C was higher than that at 500 °C.

### 3.4. Discussion

During the overlaying process, the weld pool’s solidification and cooling speeds would be very fast. Hence, the overlaying process should typically be non-equilibrium solidification and crystallization process, helpful for the formation of fine crystal structures and restraining heterogeneous segregation. Meanwhile, the high entropy thermodynamics of HEAs reduced the Gibbs free energy and inhibited the formation of complex intermetallic compound phases. In addition, the slow diffusion effect dynamics led to a low diffusion coefficient and high activation energy of the alloys. These factors contributed to the formation of a single BCC solid solution phase in the overlaying formed layer.

During the solidification of overlaying layer, the liquid phase showed a large positive temperature gradient at the bonding interface between the overlaying layer and substrate. Besides, the component undercooling degree was very small and heat propagated through the substrate. Therefore, grains at the interface were mainly grown as plane crystals. As the solid–liquid interface further formed, the temperature gradient in the liquid phase gradually declined, component supercooling degree enhanced, and grain growth turned into cellular crystals. During the growth of cellular crystals, solute atoms were discharged to the surrounding area. As crystal growth continued, the extension length to the liquid metal increased continuously, and component supercooling took place at the dendrite tip in all directions. This resulted in the growth of secondary dendrite and the formation of columnar dendrite organization (Figure 3a). However, the directionality of the heat flow weakened in the middle and top parts of the overlaying layer. Meanwhile, the component supercooling zone extended to the liquid metal with the growth of the columnar dendrites at a component undercooling degree greater than that of heterogeneous nucleation in the melt. This led to the formation of large numbers of nucleations, grown freely without any directionality, inducing finally the “cauliflower” dendrite and equiaxed crystal structure shown in Figure 3b. In addition, the stirring effect of electric arc on the molten pool promoted the diffusion of solute in the molten pool during the overlaying process, thereby restraining the segregation phenomenon. In addition, the temperature field in the molten pool was more uniform, leading to improved homogeneity and uniformity of the formed structure. On the other hand, the increase in fluidity led to re-melting and fracture of dendrite structure in the solidified area. These large amounts of broken dendrites acted as nucleation and mass points, refining the grains and promoting the development of the upper and middle zones into the equiaxed structure.

The appearance of dendrite segregation in the overlaying formed layer of WNbMoTaTi refractory HEAs was related to the melting point of elements and mixed enthalpy between them (Table 2). In addition, the negative mixing enthalpy between W-Nb and W-Ta was the largest, indicating the strongest binding force between both. By comparison, the mixing enthalpy between Ti-Ta was positive, suggesting the weakest binding force. In the non-equilibrium solidification process of solid solutions, high melting point elements should take the lead in solidification to form dendrite arms. On the other hand, low melting point elements would be excluded from dendrites to form intergranular solidification. Since the melting points of W and Ta were high, the mixing enthalpy was negative, and the binding force was strong, so they were mainly concentrated in the dendrite. By comparison, the low melting point of Ti (only 1943k) and its weak binding force with Ta resulted in their concentration in the intergranular region. On the other hand, because the melting points of Mo and Nb were higher than that of Ti but lower than those of W and Ta, their distributions were relatively uniform between dendrite and intergranular regions.

For a single disordered solid solution phase, the strengthening mechanism of the alloy was identified as solution strengthening. Therefore, the high hardness and compressive strength of the overlaying formed layers of WNbMoTaTi refractory HEAs were mainly attributed to solution strengthening. For WNbMoTaTi refractory HEAs, atoms with different sizes and properties would interact with each other, resulting in distorted BCC lattice or the so-called lattice distortion effect. This, in turn, led to the formation of a local elastic stress field, hindering the movement of dislocation to a certain extent and resulting in the alloys high strength. Meanwhile, the rapid cooling non-equilibrium crystallization of the overlaying formed layer yielded a much smaller grain size than those obtained by traditional forming methods. When the material was under stress, many fine grains boundaries acted as pins to dislocate and affect fine grain strengthening. In addition, the rapid cooling non-equilibrium crystallization would naturally lead to high-density dislocation of the layer-formed layers, aggravating the lattice distortion effect of HEAs. The high-density dislocation would also entangle each other during the formed deformation of the material, thereby improving the deformation energy and both the hardness and strength of the overlaying formed layers.

The large difference in atomic size led to serious lattice distortion in overlaying formed layers coupled with large fluctuation in lattice potential energy, resulting in alloys with low diffusion coefficients and high activation energies. The serious lattice distortion reduced the sensitivity of some properties to temperature changes [29]. The alloys maintained their single-phase solid solution structures and high-temperature structural stability at high temperatures due to their low diffusion coefficients and high activation energies.

The high-temperature strength of overlaying formed WNbMoTaTi refractory HEAs led to high-temperature structural stability. During compression at 1000 °C, the deformation can be divided into three stages. The first consisted of work hardening, in which stress continuously increased with compression since dislocation density in the alloy enhanced to form large numbers of sub-grain boundaries. This, in turn, induced numerous dislocations stacked in the material, resulting in work hardening. In the second stage of stress-strain, further increase in strain transferred the deformation energy to the environment during compression due to the high ambient temperature. Moreover, the accumulated energy inside the material rose, promoting the movement of dislocations, vacancies, or grain boundaries in the alloy. Hence, dynamic recovery and dynamic recrystallization of the material took place, explaining the dynamic softening behavior of the material.

Consequently, the stress reduced then stabilized when the dynamic softening mechanism dominated. Upon completing dynamic recrystallization, the work hardening behavior continued to strengthen under further compression, and stress enhanced with strain. The increase in stress to a certain value led to an obvious instantaneous decline followed by an increase until the breaking point of the material (Figure 8b). This may be due to the formation of a new slip system that started to move under the combined action of stress and high energy as internal dislocations accumulated during compression. At 500 °C, deformation also increased the dislocation density and led to the formation of numerous sub-grain boundaries in the alloy, leading to several dislocations stacked in the material. This, in turn, caused plug-up and produce work hardening. However, the accumulation of internal energy during compression deformation was not significant compared to that at 1000 °C. In addition, the effects of accumulated energy on dislocation, vacancy, and grain boundary movement in the alloy were not enough. The material’s dynamic recovery and recrystallization behaviors were not that obvious, meaning that work hardening played a dominant role in the whole compression deformation process, and the dynamic softening mechanism was not obvious. Therefore, the compression stress-strain curve before cracking appeared relatively straight without obvious fluctuation (Figure 8a).

## 4. Conclusions

Additive manufacturing of MoNbTaWTi as high-temperature resistant HEA with a nominal composition of Mo30.7Nb13.4Ta13.4W15.2Ti27.3 was accomplished by arc surfacing technology. The microstructures and mechanical properties of the as-formed layers were analyzed. The following conclusions could be drawn:(1)Cable-type welding wire (CTWW) is an effective way to solve the bottleneck that engineering preparation of HEAs wire is impossible to realize using traditional methods, and wire is helpful to achieve high efficiency, high quality and low-cost preparation of HEAs.(2)The overlaying formed layer of WNbMoTaTi showed a single disordered BCC solid solution phase structure with no obvious component segregation, but certain dendrite segregation occurred. The bottom structure of the overlaying formed layer was identified as a columnar cellular structure, and the upper part was a “cauliflower-like” fine dendrite and equiaxed crystal structure.(3)The serious lattice distortion and slow diffusion effect led to the high-temperature structural stability of the overlaying formed layer of WNbMoTaTi, and the single-phase BCC structure was preserved from 0 to 1400 °C.(4)The average hardness of the overlaying formed layer of WNbMoTaTi at room temperature was estimated to be about 533 HV_0.2_. At 1000 °C, the hardness fluctuated around 110 HV_1_, close to that of Inconel 718 alloy 125 HV_1_. The compressive strength was 602 MPa, much higher than that of Inconel 718 alloy (200 MPa) under the same conditions.(5)As the temperature increased from 500 to 1000 °C, the compressive strength of the overlaying formed layer of WNbMoTaTi changed from 629 MPa to 602 MPa, equivalent to only 27 MPa decrease. However, the fracture strain increased from 2.1 to 4.9%, respectively. High energy promoted the dynamic softening behavior and starting of dislocation slip system during compression deformation. The latter was the main reason why the overlaying formed layer of WNbMoTaTi maintained high strength at 1000 °C with greatly increased fracture.

In sum, these findings look promising, and future work will focus on obtaining WNbMoTaTi HEAs structures with large sizes by surfacing technology.

## Figures and Tables

**Figure 1 materials-14-04512-f001:**
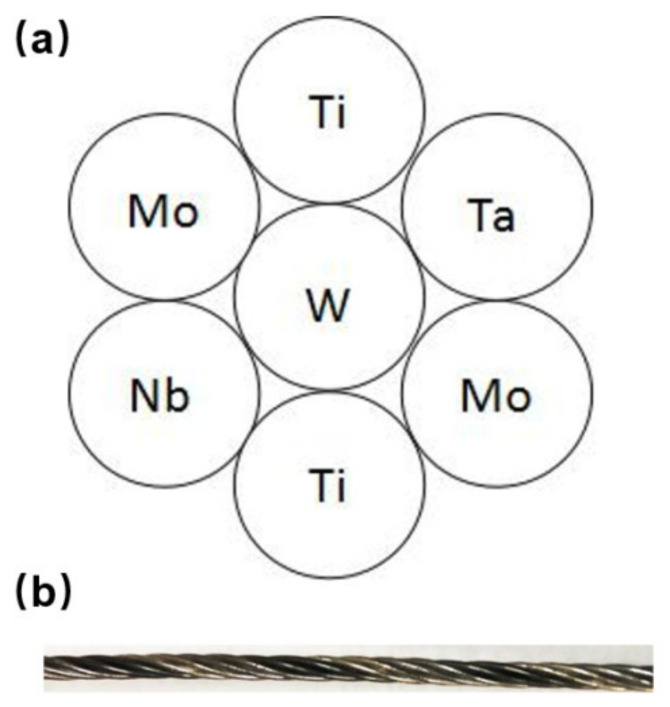
(**a**) Schematic diagram and (**b**) prepared graph of MoNbTaWTi welding wire.

**Figure 2 materials-14-04512-f002:**
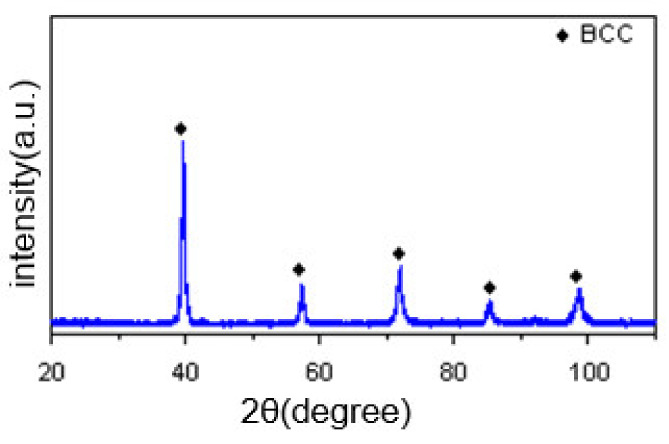
XRD patterns of the Arc Surfacing layer of MoNbTaWTi HEA.

**Figure 3 materials-14-04512-f003:**
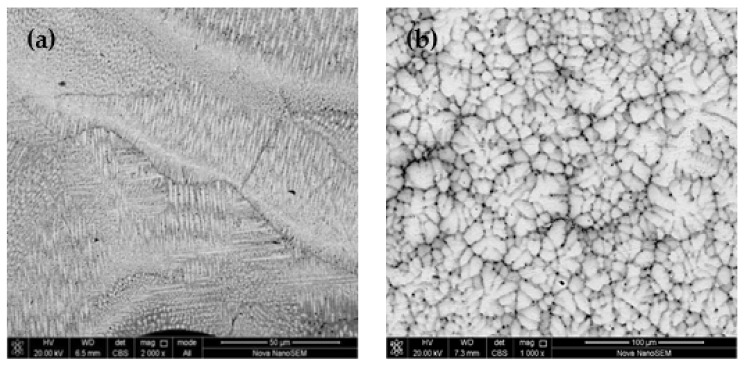
SEM image of MoNbTaWTi Arc surfacing layer: (**a**) bottom layer and (**b**) middle layer.

**Figure 4 materials-14-04512-f004:**
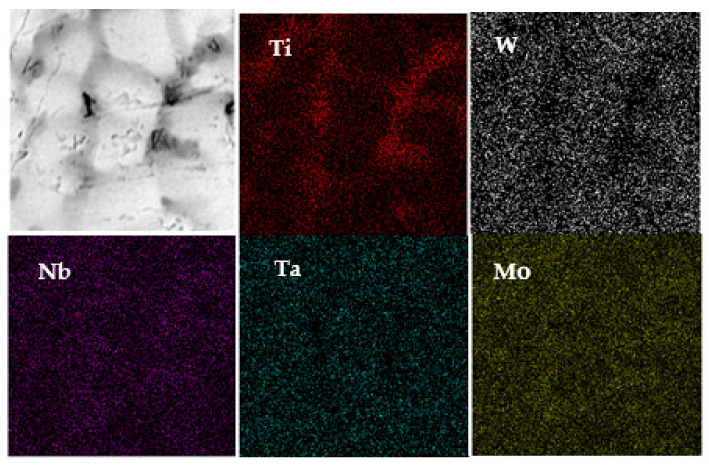
Area-scan distributions of elements in WNbMoTaTi layer.

**Figure 5 materials-14-04512-f005:**
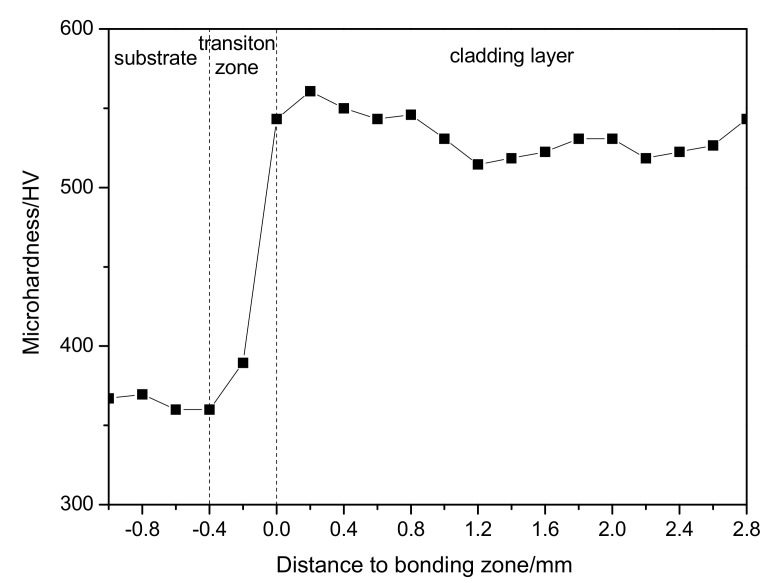
Microhardness of MoNbTaWTi Arc Surfacing layer.

**Figure 6 materials-14-04512-f006:**
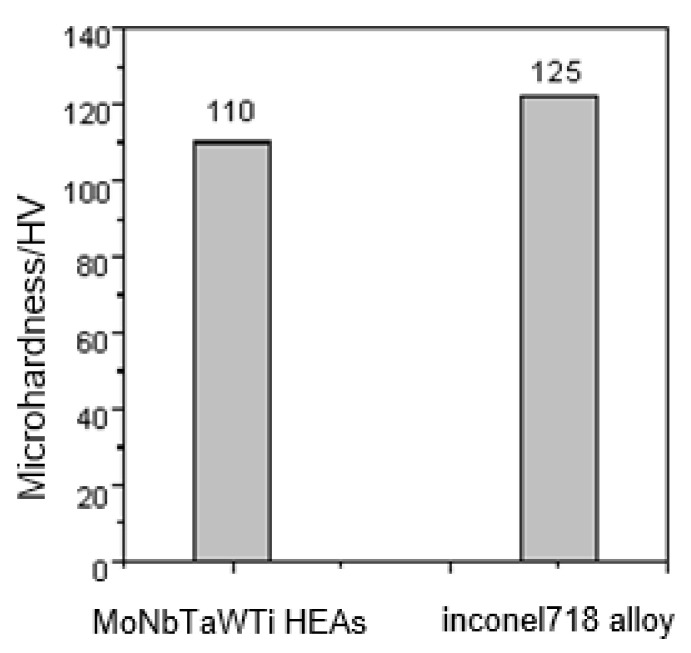
Microhardness of MoNbTaWTi Arc Surfacing layer and Inconel718 alloy at 1000 °C.

**Figure 7 materials-14-04512-f007:**
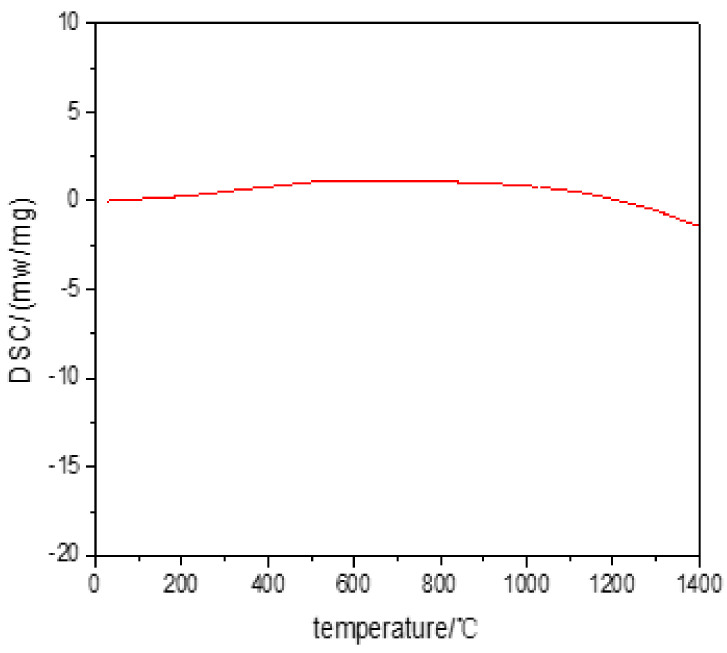
DSC curve of MoNbTaWTi Arc Surfacing layer.

**Figure 8 materials-14-04512-f008:**
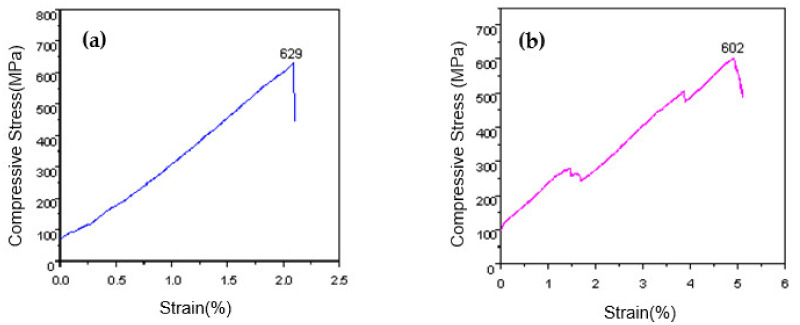
Compressive stress-strain curves of MoNbTaWTi Arc Surfacing layer at the high temperatures: (**a**) T = 500 °C and (**b**) T = 1000 °C.

**Figure 9 materials-14-04512-f009:**
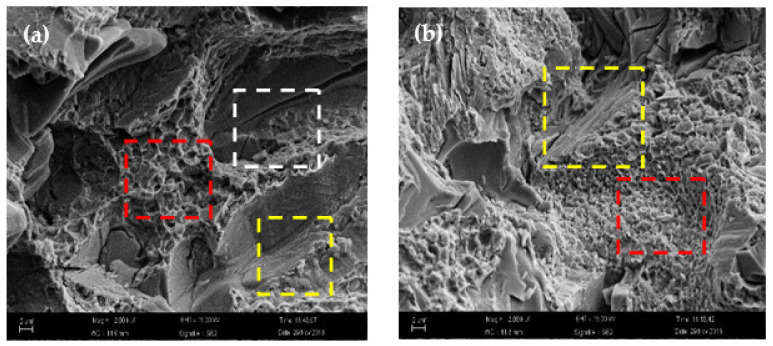
Fracture morphology of MoNbTaWTi Arc Surfacing layer after compressive deformation at high temperatures: (**a**) T = 500 °C and (**b**) T = 1000 °C.

**Table 1 materials-14-04512-t001:** Chemical composition of WNbMoTaTi HEAs layer (at%).

		W	Nb	Mo	Ta	Ti
Atomic percentage	Dendrites (white color)	20.88	12.87	33.87	14.95	17.63
	Intergranular (dark color)	8.45	15.31	30.64	9.08	36.52
Segregation rate SR		0.40	1.19	0.90	0.61	2.07

**Table 2 materials-14-04512-t002:** Mixed enthalpy between elements (KJ/mol).

Element	W	Nb	Mo	Ta	Ti
W	0	−8	0	−7	6
Nb	-	0	−6	0	2
Mo	-	-	0	−5	−4
Ta	-	-	-	0	1
Ti	-	-	-	-	0

## Data Availability

The raw/processed data required to reproduce these findings cannot. be shared at this time as the data also forms part of an ongoing study.

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
