# Peer review of "Microstructure and Mechanical Properties of Wire Arc Additively Manufactured MoNbTaWTi High Entropy Alloys"

_materials, 2021, doi:10.3390/ma14164512_

Round 1

Reviewer 1 Report

The manuscript presents a work on microstructure and mechanical properties of high entropy alloys. 

The work cannot be considered for publication in the Materials. Authors must make significant changes for the manuscript to be of scientific soundness in order to be published.

The introduction should be improved in order to show a solid brackground showing the innovation and motivation of this particular study. A two-paragraph introduction is not acceptable for a paper in a journal such as Materials.

The experimental procedure section does not present complete information on the production and characterization of the alloys under study.

The presentation and discussion of results is also far from expected for a scientific article to be published. Many of the results presented in the form of images and graphics are not very careful in their presentation. There is a need for greater quality in the results to be introduced. It is necessary to better the scientific part of this work

Reviewer 2 Report

The paper about arc surfacing AM technology, with quite interesting and self-made material like MoNbTaWti steel is very interesting. Material properties tested at different temperatures and compared to Inconel 718 are a good idea and were properly thought through and executed.

What I found insufficient was the introduction. I believe adding a little more information about the technology and its impact on tested properties, in general, would be a good place to start. Also, the experimental part seems ‘rushed’. Even though every part of the experiment is described here, with the equipment and in a very composed way, it somehow is hard to read and comprehend everything that was done by the Authors. I suggest expanding this part, adding some more details and maybe a few pictures or drawings.

Also, the formatting of the paper I was given is lacking. For example, figures 2, 6 and 7 are poorly aligned with the text, same goes for tables 1 and 2.

Figure 3 could be bigger, it's hard to read.

Having said all that, I recommend this paper for publication. After minor revision of course.

Regards

Reviewer 3 Report

In this article, the authors explore some mechanical properties of MoNbTaWTi alloy after bring processed using the arc surfacing AM process. The study reported is interesting, provides some useful new perspective for the literature, and is relatively novel for the literature. This type of AM process is under-studied in my opinion and I am glad that the authors decided to focus a study on it. However, the manuscript itself is very far from acceptable for publication in its current form. The quality of the language is generally good, but large pieces of what would normally be expected in a paper like this are missing. Please see my detailed comments below. I would normally like to see more detailed and diverse physical experiments in a characterization paper, but given the unusual process and material studied, I think there is minimally sufficient new data for publication.

Given that I believe the study itself is sufficient to provide some new information in the literature and that most of my issues have to do with the manuscript itself, I recommend that the paper be sent back to the authors for a major revision followed by another round of review. 

Comments: 

1. The abstract is not useful, as it really does not address the actual purpose of the study, the approach, the major conclusions, etc. 

2. In my opinion, the title seems to be promising too much that the paper does not deliver. As I said in my previous comments, I consider this basically a screening study or preliminary findings reported to the community. There is enough novelty to justify publication given the process and material, but it is very short and very superficial compared to many of the very rigorous characterization papers that already exist in the literature for other processes and materials. The authors should consider changing the title to something that is more realistic relative to the actual content of the paper. 

3. The introduction and literature review section is practically non-existent. There is very little background provided, no discussion of novelty or of issues in the literature that need to be addressed, and no serious analysis of previous work. The whole section is less than half a page. This is not acceptable, as a detailed and rigorous intro/literature review is necessary to show the value of doing and publishing a new study. Under no circumstances would I recommend this paper for publication until a very good section was written to address this point. 

4. The materials and methods section is extremely poor and is not sufficient to be able to reproduce the experiments described. This section is very important and needs to be very detailed and clear. If this section was turned in as part of a lab report for one of my undergraduate classes, it would receive a failing grade. Naming equipment and listing some parameters is not nearly enough information to be able to reproduce the experiments. Once this section is done properly, I will be able to better evaluate the data and conclusions presented so please expect more comments on this section during the section review if I am asked to review a revision.  

5. The authors used "self-made" feed wire for the deposition process. The paper needs to provide a lot more detail about this, enough detail for a reader to be able to made and use the same wire given access to the standard needed equipment. 

6. The analysis of the surface characteristics is ok, but the analysis of the mechanical properties needs to be described in more detail. More figures would help. Check some of the existing figures (Figure 5-6 in particular) to make sure they are clean  and clear and that residual non-English characters are not present. 

7. The discussion section is actually pretty good. It could do with some more polishing and discussion relative to the new literature review, however. 

8. Some of the conclusions are only weakly supported by the data shown in this paper, especially considering that the collected dataset is small, does not use any kind of experimental design, and is not meaningfully replicated within the paper. Therefore, the authors should review the conclusions and make sure that all of them are truly supported by the available evidence in the literature and data from the study. 

9. The author contributions statement is missing. Given the small size and superficial paper presented, there seems to be an unusually large number of authors. Check that each author actually meets the criteria for authorship. Remember that funding the research group, being the lab head, reviewing drafts, supervising the project, and providing advice/feedback are not in themselves enough to qualify an author. 

10. The affiliation of most of the authors is to a military university. If any of the authors have direct/official roles (including any military rank) with a government or military force or company affiliated with any military force, this needs to be clearly disclosed during the conflict of interests statement for the paper. Please note that this comment is not pointed at the specific country of origin of this paper and is true for any nation in the world. 

11. Finally, personal email addresses (the corresponding author has an @163.com address) are not acceptable for scholarly papers. A stable, official email address should always be used. 

Good luck with the revision and I look forward to potentially seeing the next version of the paper in a few weeks. 

Round 2

Reviewer 1 Report

The authors made changes in accordance with some reviewers' comments.

Author Response

dear sir,thank you again for the review and feedbacks.

The editor only gave us three days to revise the manuscript. Maybe we need discuss it with the editor to obtain enough time to revise the manuscript well. We will try our best to submit a perfect manuscript.

Wish you all the best.

Reviewer 3 Report

Upon reviewing the revised manuscript, my major concerns with technical content have been addressed. I do however feel that the revision could have been stronger and more thorough. I will leave it to the editor to decide if the answers to the technical questions are sufficient. I think they are fairly superficial but good enough to go forward with given the novelty of the study. Therefore I recommend acceptance with reservations - see below for details of this. 

I do want to make two comments in response to the authors' reply letter for the editor to consider. 

1. Being affiliated with a military university (in any country) does not exempt the authors from the normal expectations of quality scholarship that we are all expected to follow - declining to provide some information based on this is not good practice for an international journal. In addition to being affiliated with a military university, authors being actual officers or members of the military is a potential conflict of interest in itself. Details such as rank and security clearance level do not need to be shared, but which authors are active military and which are civilians is important information in judging the credibility and objectivity of the work, especially when we are talking about advanced engineering works. If the work is somehow protected or secret or the authors have protected identities, the paper should be published using official government protocols and classifications and not in an open-access international journal. Once again, I want to make sure the authors understand that I am not singling them or their country out by this point - I would (and have) make the exact same comment about a paper from a military-affiliated university, company, or group in any country including my own. 

2. I cannot not accept the word of anyone that a personal email address is acceptable to put on scholarly work. Unless official email addresses physically do not exist and cannot be created, they must be used. Even addresses associated with a professional society (ASME, IEEE, etc) would be better than using personal ones because at least the society would provide some kind of identity verification.  

Good luck to the authors from here - the editor will decide if my points here should be considered further or not. 

Author Response

dear sir ,thank you again for the review and feedbacks. i wish you all the best
